# Association between Vitamin D Receptor Gene Polymorphisms and Periodontal Bacteria: A Clinical Pilot Study

**DOI:** 10.3390/biom12060833

**Published:** 2022-06-15

**Authors:** Concetta Cafiero, Cristina Grippaudo, Marco Dell’Aquila, Pasquale Cimmino, Antonio D’Addona, Paolo De Angelis, Maria Pia Ottaiano, Domenico Costagliola, Giulio Benincasa, Alessandra Micera, Luigi Santacroce, Raffaele Palmirotta

**Affiliations:** 1Pathology Unit, Fabrizio Spaziani Hospital, 03100 Frosinone, Italy; 2Department of Head and Neck, Division of Oral Surgery and Implantology, Catholic University of the Sacred Heart, Fondazione Policlinico Gemelli IRCCS, 00168 Rome, Italy; cristina.grippaudo@unicatt.it (C.G.); antonio.daddona@unicatt.it (A.D.); dr.paolodeangelis@gmail.com (P.D.A.); 3Section of Anatomic Pathology, Department of Life Sciences and Public Health, Catholic University of Sacred Heart, 00168 Rome, Italy; mzrk07@gmail.com; 4Department of Oral and Maxillofacial Surgery, Federico II University, 80100 Naples, Italy; pasquale.cimmino67@gmail.com; 5Department of Clinical Pathology and Molecular Biology, Pineta Grande Hospital, 81030 Castel Volturno, Italy; mariapia.ottaiano@gmail.com (M.P.O.); domenico.costagliola@pinetagrande.it (D.C.); giulio.benincasa@pinetagrande.it (G.B.); 6Research Laboratories in Ophthalmology, IRCCS—Fondazione Bietti, 00184 Rome, Italy; 7Interdisciplinary Department of Medicine, University of Bari “Aldo Moro”, 70124 Bari, Italy; luigi.santacroce@uniba.it (L.S.); raffaele.palmirotta@uniba.it (R.P.)

**Keywords:** VDR gene, genetic polymorphism, periodontal pathogens, periodontitis, haplotype analysis, genotype association

## Abstract

**Background:** Periodontitis is an inflammatory disease caused by microorganisms involving the supporting tissues of the teeth. Gene variants may influence both the composition of the biofilm in the oral cavity and the host response. The objective of the study was to investigate the potential correlations between the disease susceptibility, the presence and the quantity of periodontopathogenic oral bacterial composition and the VDR gene polymorphisms. **Methods:** Fifty (50) unrelated periodontal patients and forty-one (41) healthy controls were selected for genomic DNA extraction. DNA concentration was measured and analyzed. The periodontopathogenic bacterial species were identified and quantified using a Real Time PCR performed with species-specific primers and probes. **Results:** Genotype distribution showed a different distribution between the groups for BsmI rs1544410 genotypes (*p* = 0.0001) with a prevalence of the G(b) allele in periodontal patients (*p* = 0.0003). Statistical significance was also found for VDR TaqI rs731236 (*p* ≤ 0.00001) with a prevalence of the T(T) allele in periodontal patients (*p* ≤ 0.00001). The average bacterial copy count for the periodontitis group was significantly higher than that of control group. Dividing patients into two groups based on high or low bacterial load, FokI rs2228570 T allele (f) was statistically more represented in patients with high bacterial load. **Conclusions:** The findings of the study suggest the involvement of the VDR gene BsmI and TaqI polymorphisms in periodontal disease, while FokI and BsmI may be involved in determining an increased presence of periodontopathogens.

## 1. Introduction

Periodontitis is an inflammatory disease caused by specific microorganisms involving the supporting tissues of the teeth, resulting in progressive destruction of the periodontal ligament and alveolar bone with the formation of pockets, featuring gingival bleeding, pathological tooth mobility and abscesses, and ending with teeth loss [1,2,3]. To the best of our knowledge to date, periodontitis is mediated by a complex interaction between dysbiosis of the oral microbiota and an aberrant immune response of periodontal tissues [4]. The oral microbiome consists of about 700 species of bacteria, characterized by a complex network of metabolic and physical interactions [5,6]. When the oral biofilm complex is disrupted and the bacterial load increases, microbial dysbiosis arises, the irritative response of host tissues increases and epithelial cells produce pro-inflammatory cytokines and other inflammatory mediators that contribute to the development of periodontal disease [7,8,9,10]. Several bacterial species associated with the development of periodontitis were initially defined by Socransky et al. that identified *Porphyromonas gingivalis, Tannerella forsythia*, and *Treponema denticola* as species associated with severe periodontal disease, thus coining the cluster known as “the red complex” [6,9,10,11]. Over the years, however, periodontitis-associated species have expanded beyond the red complex to include numerous other bacterial phyla associated with the disease [12,13,14]. In addition to assessing the oral microbial species as causative of periodontitis, the scientific community has focused on other concomitant mechanisms implicated in the pathogenesis of the disease. The main cause of periodontal disease is the development of the biofilm on the teeth due to lack of oral hygiene. Several risk factors contribute to the development of periodontal disease such as obesity, diabetes, inadequate nutrition, vitamin C deficiency, hormonal changes, drug use, smoking, systemic conditions characterized by decreased immune function and genetic factors [15,16,17]. The role of risk factors for the development of periodontal disease should be carefully considered because they can change the susceptibility or resistance of patients to the inflammation and the disease. Therapeutic interventions aiming to modulate the inflammation profile should include behavioral interventions, such as smoking and dietary consumption of calcium and vitamin D and the treatment of medical conditions (e.g., poorly controlled diabetes, stress, and osteopenia). Recently, obesity and low levels of physical activity were also classified as risk factors for periodontitis [18]. Among the innovative approaches for treating inflammatory conditions is also Hyperbaric oxygen (HBO) therapy which has been recently introduced and investigated for this clinical scenario with promising results [19]. Conversely, numerous studies have highlighted the importance of oral health considering periodontal disease as an aggravating factor in the course of systemic diseases such as diabetes, cardiovascular diseases, and autoimmune disorders as well as Alzheimer’s disease, oral cancer, and inflammatory bowel diseases [2,16,17,20]. Regarding the genetic factors, the massive developments in molecular biology in the last decades have allowed to identify gene variants that in combination with lifestyle and environmental factors influence both the composition of the biofilm in the oral cavity and the host inflammatory immune response [8,21]. Research on genetic factors has focused primarily on genes modulating the immune system, such as genes encoding for the cytokines IL1A, IL1B, IL10, and IL6, considered key factors in the inflammatory process during periodontal disease [8,22,23]. Numerous studies have also been performed on vitamin D receptor (VDR) encoding gene polymorphisms BsmI, TaqI, FokI, and ApaI, opening to new considerations [24,25,26,27,28,29,30,31]. Once activated, the VDR protein modulates the transcription of genes that promote the functions of vitamin D, and it is established that its sequence variants are associated with dysfunctions in the metabolism of vitamin D [29,30]. These SNPs have been considered a key factor in the reabsorption of alveolar bone and increased bone cell turnover resulting in an increased risk of reduced bone mass density (BMD) and osteoporosis [32]. However, despite the large number of studies on VDR polymorphisms, it has not yet been clearly established whether any of them can influence the risk of periodontal disease, and in many cases, the results are contradictory and controversial [28]. In addition, few studies have attempted to correlate the presence and levels of periodontal pathogens, which are often altered in a preclinical phase of the disease, with a particular host genetic predisposition [21,33,34,35,36]. However, several studies would seem to suggest that the role of the Vitamin D would be that of preserving alveolar bone density, with polymorfisms of its receptor being a risk factor for alveolar bone loss, thus increasing the risk of periodontal diseases [37,38,39]. Moreover, it has been demonstrated in vivo the role of Vitamin D hydroxylases as a factor implicated in a reduced Vitamin D activity [40]. Studies both in vitro and on animal models demonstrated the role of Vitamin D in periodontal disease by inducing the expression of antimicrobial peptides and innate immune mediators in gingival epithelial cells, thus enhancing innate immune defenses against microbes [41]. Interestingly, a role in this context has also been found for the Vitamin D binding protein, as an active actor in this complex interplay [42]. Based on these findings, a pilot study was performed analyzing the VDR polymorphisms FokI, BsmI, ApaI, and TaqI on a selected Italian population composed of fifty (50) patients with periodontal disease and forty-one (41) healthy controls. The objective was to investigate both a possible susceptibility to the disease and a potential association between the oral bacterial composition and the VDR gene polymorphisms of the patients. For this latter purpose, the investigation has been focused on *Aggregatibacter actinomycetemcomitans. Porphyromonas gingivalis, Porphyromonas endodontalis, Treponema denticola, Tannerella forsythia, Prevotella intermedia*, and *Fusobacterium nucleatum* by comparing bacterial loads with genotypes identified in patients.

## 2. Materials and Methods

### 2.1. Population Study

Based on the compliance with the study, after informed consent subscription, we selected 50 unrelated Caucasian patients (26 males and 24 females; mean age 47 ± 8 yrs; range 21–63 yrs) affected by periodontal disease treated at the Dental Clinic of the Policlinico “A. Gemelli” (Rome, Italy) between 2019 and 2020. All patients were older than 18 years. We applied the following exclusion criteria: medically compromised patients, treatment with antibiotics or antimicrobials in the past 6 months, and pregnant and lactating women. Periodontal disease was diagnosed in accordance with the case definitions by the American Academy of Periodontology (AAP) [3]. As controls, 41 unrelated healthy subjects (19 males and 22 females; mean age 44 ± 11 yrs; range 26–65 yrs) that matched with the patient’s group for age, gender, and ethnicity were enrolled. The study protocol was prepared in accordance with the Declaration of Helsinki and Ethical approval was obtained from the Committee of the Catholic University of Sacred Heart, Roma (#UCSC prot. 36110/10 ID: 565). Written informed consent was obtained from each subject before participating in the study and biological sampling.

### 2.2. Sampling and DNA Extraction

In all subjects, the biological samples were obtained from the deepest periodontal pocket in each quadrant of the dentition by using sterile paper points [43]. The tips of paper left inside the periodontal pocket for 30 s were then inserted into a 1.5 mL sterile tubes with 300 μL of sterile phosphate buffer and transferred to −80 °C freezer until further DNA extraction. Specimens included periodontal microflora but also enough host cells that enabled genetic profiling of patients and healthy controls. Genomic DNA was extracted using the Ampli DNA EXTRA Kit (Dia-Chem Srl, Molecular Biology, Naples, Italy) according to the manufacturers’ protocol. DNA concentration was measured with NanoDrop (Thermo Fisher Scientific, Waltham, MA, USA), and samples with A260/280 ≥1.8 were considered suitable for further analysis [44].

### 2.3. Identification and Quantification of Periodontal Pathogens

The periodontopathogenic bacterial species subject of this study were identified and quantified using a Real Time PCR performed with species-specific primers and probes for *Aggregatibacter actinomycetemcomitans*, *Porphyromonas gingivalis*, *Porphyromonas endodontalis*, *Treponema denticola*, *Tannerella forsythia*, and *Prevotella intermedia* (Appendix A). Primer and probe sets were designed from the variable regions of the 16S rRNA gene sequences obtained from the Ribosomal Database Project release 10 [45] using Primer Express Software Version 10 (Thermo Fisher) Selected primers, and probes were checked for homology with unrelated sequences with the Basic Local Alignment Search Tool (BLAST) (https://blast.ncbi.nlm.nih.gov/Blast.cgi (accessed on 3 January 2022)) [46] (Table 1).

Before carrying out quantitative analyses, cloned plasmids containing the amplified region of each target bacterium were obtained using the TOPO™ XL-2 Complete PCR Cloning Kit with linearized and topoisomerase 1-activated pCR™-XL-2-TOPO™ vector (Invitrogen—Thermo Fisher). The obtained plasmids were purified using MaxiPrep (Qiagen) and quantified with NanoDrop (Thermo Fisher Scientific, Waltham, MA, USA) spectrophotometry at multiple dilutions. Thus, a standard curve was generated using quantified plasmid standards containing the target DNA sequence using serial dilutions of 10, 102 to 108 plasmid copies. Standard plasmid and clinical samples were analyzed in duplicate, and the mean values were used for the calculation of the bacterial load. Assays were performed in a volume of 35 µL containing 17.5 µL TaqMan Universal PCR Master, 7.5 µL of extracted DNA, and 10 µL of PCR mix consisting of 1.5 µL of MgCl2, dNTP mix, 20 pmol of forward and reverse primers (Eurofins), and 4.0 pmol of Taq Man probes (Thermo Fisher).The amplification was performed using the following conditions: 80 °C for 2 min 1 cycle, 95 °C for 90 s 1 cycle, followed by 95 °C 15 s, 60 °C 30 s, and 72 °C 40 s for 40 cycles in a cycler CFX96 Touch Real-Time PCR Detection System (Biorad). The results of the quantitative analysis are expressed in copies/mL.

### 2.4. Determination of VDR Gene Polymorphisms

The FokI (rs2228570) BsmI (rs1544410) ApaI (rs7975232) and TaqI (rs731236) VDR polymorphisms were analyzed on DNA from patients and controls using the commercial kit AMPLI set VDR Polymorphisms (Dia-Chem Srl, Molecular Biology, Naples, Italy) according to the manufacturer’s procedures. The kit first requires a PCR amplification with specific primers for the gene regions containing the polymorphisms and then a DNA enzyme restriction assay. The restriction products are subsequently electrophoretically separated on ethidium bromide-stained 4% agarose gel to identify the various genotypes for each polymorphism. In this regard, it is appropriate to specify that conventionally the nomenclature of alleles was initially determined on the basis of the presence or absence of the restriction sites, using small or capital letters respectively [47]. Considering the heterogeneity of the allele definition literature in order to simplify the reading and interpretation of the results, in our study, we identified the various alleles with a double nomenclature indicating both nucleotides and conventional nomenclature (in brackets). Thus, in genotype identification, FokI C(F) T(f), BsmI A(B) G(b), ApaI A(A) C(a), and TaqI T(T) C(t) are described.

### 2.5. Statistical Analysis

Allelic frequencies (%) were estimated by gene counting, and genotypes were scored. The observed frequencies of each SNPs genotype were compared with those expected for a population in Hardy–Weinberg equilibrium (HWE). A comparison between the genotyping of our four analyzed VDR SNPs and the allele frequency data available from the Genome Aggregation Database v3.1 (GnomAD, https://gnomad.broadinstitute.org (accessed on 30 March 2022)), the 1000 Genomes Browsers (IGSR: The International Genome Sample Resource, https://www.internationalgenome.org (accessed on 30 March 2022)), and the Ensembl project (https://www.ensembl.org (accessed on 30 March 2022)) was performed [48,49]. The significance of the differences of observed genotypes and alleles, haplotype frequencies, linkage disequilibrium, and associations between groups as well as analysis of multiple inheritance models (codominant, dominant, recessive, and over-dominant) were verified using free web-based applications SNPStats software (http://bioinfo.iconcologia.net/snpstats/start.htm (accessed on 16 March 2022)) and SHEsis software (http://analysis.bio-x.cn/myAnalysis.php (accessed on 16 March 2022)) [50,51,52]. A formal sample size calculation was performed at beginning with the following parameters: effect size d = 0.5 (medium); α error probability = 0.05 and an allocation ratio of 1 (Genetic Power Calculator G Power 3.1.9.4; free available) [53]. The real power of this study population, for a total size of 91 subjects, was 0.76, considering means from these two independent groups (50 case and 41 controls). Row values were analyzed using the Shapiro-Wilk test and F test to satisfy normality and variance assumptions. Data were analyzed using Student’s *t*-test or one-way ANOVA with Bonferroni post-test, as appropriate. Two-sided tests were used for analysis, and *p*-values lower than 0.05 were regarded as statistically significant. All statistical analysis was performed using GraphPad Prism 5 software (GraphPad Software, La Jolla, CA, USA; https://www.graphpad.com (accessed on 18 March 2022)).

## 3. Results

### 3.1. Comparison of Allelic Frequencies with Population Databases

Our case-control study included 91 Caucasian individuals, subdivided in 50 periodontitis patients and 41 healthy controls. We first compared the allele frequencies observed in our study with those reported in in the 1000 Genomes Project Phase 3 and GnomAD genomes v3.1, extrapolating in both cases Global and European population databases (Table 2). BsmI rs1544410 SNP displayed a significant different frequency distribution in our population with respect to European populations from both databases (*p* = 0.001). TaqI rs731236 showed significant difference with the European Population from the 1000 Genome Project (*p* = 0.002) as well as with Global and European populations from GnomAD genomes (*p* = 0.002) (Table 2).

### 3.2. Genotype Association Analysis

Genotypes and allele frequencies observed in health controls and periodontitis patients are reported in Table 3.

Among all patients and control subjects’ groups, the genetic distributions of analyzed SNPs did not deviate from the Hardy–Weinberg equilibrium, and no significant differences were found between genotypes frequencies and gender or age in all subjects. Genotype distribution showed a different distribution between the two groups for BsmI rs1544410 genotypes (*p* = 0.0001) with a prevalence of the G(b) allele in patients (*p* = 0.0003). The A/G(B/b) genotype in the control health group was more frequent in codominant [OR (95% CI): 0.16 (0.06–0.43), *p* = 0.0003], dominant A/G-A/A (B/b-b/b) [OR (95% CI): 0.16 (0.06–0.40), *p* = 0.0001] and overdominant [OR (95% CI): 0.18 (0.07–0.48), *p* = 0.0003] inheritance models. Statistical significance was also found by comparing genotypic frequencies between the two groups for VDR TaqI rs731236 (*p* ≤ 0.00001) with a prevalence of the T(T) allele in patients (*p* ≤ 0.00001). In this case, the T/C (T/t) genotype was found with higher frequency in in the control health group in codominant [OR (95% CI): 0.08 (0.03–0.25), *p* ≤ 0.0001], dominant T/C-C/C (T/t-t/t) [OR (95% CI): 0.06 (0.02–0.20), *p* ≤ 0.0001] and overdominant [OR (95% CI): 0.11 (0.03–0.32), *p* ≤ 0.0001] inheritance models. In addition, comparing the VDR ApaI rs7975232 allele frequencies obtained between controls and patients, we found a significant difference finding allele A (A) more frequent in the control group (*p* = 0.026), while no difference was present when comparing genotypic frequencies (*p* = 0.064).

### 3.3. Haplotype Analysis

Haplotype analysis performed using the four selected SNPs, considering a minimum frequency in either group of at least 3%, demonstrated the occurrence of 11 haplotypes.

In particular, the haplotype FokI, BsmI, ApaI, and TaqI TGCT (fbaT) was exclusively present in the group of patients (28.8%) (*p* = 0.0000000936). Conversely, haplotypes CGAC (FbAt), TACT (fBAt), TAAT (fBAT), and CGCC (Fbat) were found only in the control group with frequencies of 10.9% (*p* = 0.001302), 10.0% (*p* = 0.001335), 4.9% (*p* = 0.026577), and 10.4% (*p* = 0.001051), respectively. The prevalence of the other haplotypes was comparable between the two groups (Table 4). Finally, Levontin’s standardized disequilibrium coefficient (D’), calculated as a measure for LD among investigated SNPs in the VDR gene, showed a moderate LD between ApaI rs7975232 and TaqI rs731236 (D’ = 0.545, r2 = 0.095) (Table 5).

### 3.4. Bacterial Load Assessment

The amount of the bacterial load by real-time PCR from health controls and periodontitis patients’ samples, expressed as number of copies/mL, is reported in Table 6. The average copy count for the periodontitis group was significantly higher than that of control group and the highest value was observed for *P. gingivalis* and *P. endodontalis*, with almost similar averages of 3.963.445 and 3.962.550 copies/mL, respectively. Furthermore, *P. intermedia* and *F. nucleatum* had lower average values of 1.466.116 and 936.526, while *A. actinomycetemcomitans, T. denticola*, and *T. forsythia* presented values of 264.331, 373.472, and 363.527 copies/mL, respectively. The most prevalent specie in health controls was *F. nucleatum* with an average of 13.540 copies/mL. Specific values for each patient and control subject are detailed in Appendix A. We then assigned an arbitrary cut off to the values found for each bacterium, in order to divide the 50 patients in the study into two defined groups, so that we obtained a consistent number of cases (HIGH bacterial load) and controls (LOW bacterial load) for subsequent analysis of correlation between bacterial load and genotypes (Table 6).

### 3.5. Genotype Association Analysis between Patients with High and Low Bacterial Load

The genotypes and the allelic frequencies did not significantly deviate from the Hardy–Weinberg equilibrium between the two groups of patients selected on the basis of a cut off of the bacterial load for each investigated microorganism. However, regarding *Fusobacterium nucleatum* the homozygous A/A (*B/B*) and the heterozygous A/G (*B/b*) BsmI variant genotypes were more frequent in patients with LOW bacterial load (*p* = 0.049) with a statistical significance in codominant, dominant, and overdominant inheritance models (Table 7). Conversely, for all other bacteria, a significant association was found between the homozygous T/T (*f/f*) and the heterozygous C/T (*F/f*) FokI variant genotypes in patients with HIGH bacterial load with a statistical significance in codominant, dominant, and overdominant inheritance models (Table 7). The genotypic and allelic distributions of VDR polymorphisms found in patients with HIGH and LOW bacterial load are detailed in the Appendix A.

### 3.6. Haplotype Analysis in Patients with High and Low Bacterial Load

Haplotypes, constructed with frequency threshold for rare haplotypes of <5%, allowed to identify 5 haplotypes that accounted 90.8% of estimated haplotypes in the 50 patients. Since the order of the haplotypes was always FokI, BsmI, ApaI, and TaqI, the TGCT (fbaT) was the most represented (28.6%) and statistically more frequent in patients with HIGH bacterial load for *P. gingivalis*, *P. endodontalis*, *T. denticola*, *T. forsythia*, and *P. intermedia*. In contrast CGCT (FbaT), which showed a frequency of 26.0%, was significantly related to patients with LOW bacterial load for *T. denticola*, *T. forsythia*, and *P. intermedia* (Appendix A).

## 4. Discussion

We analyzed the genetic variants on the VDR gene in a cohort of 50 Italian periodontal patients and 41 healthy control subjects in order to investigate both a potential correlation with the disease susceptibility and the presence and quantity of periodontopathogenic bacteria. The study is based on the hypothesis that these polymorphisms can not only be associated with the periodontal disease, but also be responsible for the qualitative and quantitative composition of the subgingival microbiota. Several authors have considered the potential impact of FokI, BsmI, ApaI, and TaqI VDR gene variants as potential genetic factors involved in susceptibility to periodontitis. However, to date the numerous studies investigating the association between these single or combined different genetic variants and the disease are controversial and contradictory. Moreover, even the meta-analysis studies carried out so far, including a high number of trials and reporting some significant associations, undoubtedly show conflicting results. The meta-analysis conducted by Deng H. et al. (2011) of 15 studies of Asian and Caucasian cohorts, including 1338 cases and 1302 controls, identified a lower frequency of GG (bb) BsmI, a higher frequency of ApsI AA (AA), and an equally higher frequency of TaqI TT (TT) genotypes exclusively in patients of Asian ethnicity [24]. Conversely, applying the same analytical methodology to nine Chinese studies, with 1014 periodontitis cases and 907 controls, Ji X.W. et al. (2016) did not identify association for the TaqI polymorphism [25]. Through a meta-analysis that considered 19 publications, Mashhadiabbas F. et al. (2018) found no association between VDR gene polymorphisms and risk of chronic periodontitis [26]. However, stratifying samples by ethnicity, it was possible to observe a significant association between the allele A (B) of the BsmI polymorphism and risk of chronic periodontitis only in the Caucasian subgroup [26]. Instead, an analysis of 30 studies conducted by Yu X. et al. (2019) indicated that exclusively FokI’s C allele (F) was significantly associated with periodontitis susceptibility, with increased prevalence in East Asian ethnic groups [27]. Furthermore, Wan Q.S. et al. (2019) performed a larger meta-analysis of 34 previous studies including 3848 periodontitis patients and 3470 controls with different Asian, Caucasian, African, and Arabian ethnic groups. In this case, in the overall population, a correlation was found between periodontitis and the BsmI and FokI gene polymorphisms with a prevalence of the A allele (B) and the T allele (f), respectively. In addition, a correlation between the C (t) TaqI allele and periodontitis susceptibility was found only among the Caucasian population [28]. These discrepancies may be due both to the relatively small number of studies and cohorts considered and to differences in the frequency of variants related to the ethnicity of the populations. Regarding BsmI, for which in our analysis we found a higher frequency of the G(b) allele in patients (Table 3), the allele frequencies reported for the European population were G (b) = ~59% and A (B) = ~41% (Table 2), while those reported for East Asian populations were G (b) = ~95% and A (B) = ~5% (https://www.ensembl.org (accessed on 30 March 2022)). BsmI polymorphism is characterized by a G→A (b→B) transition in intron 8 of VDR gene (c.1024 + 283G > A). The variant does not determine structural alterations of the protein, but the presence of the G allele (b) affects the polyadenylation of the transcript, influencing the stability of mRNA and therefore modifying the protein expression. However, several meta-analyses have only shown that the G (b) allele showed lower but not significant bone mass density values [24,31]. Regarding TaqI polymorphism, our results are in line with those obtained in previous studies of Italian subjects with periodontitis in which T (T) allele were higher in patients than in controls, although in some previous works on Caucasian population the C (t) allele seems to increase the risk of developing periodontitis. The TaqI polymorphism is characterized by a T→C (T→t) transition (c.1056T > C) at exon 9 of the VDR gene resulting in a silent mutation at codon 352 (p.Ile352=). Copious evidence has established that the presence of the T allele (T) determines a reduced translational capacity and RNA stability and is associated with lower VDR mRNA levels [32]. Moreover, in this case, it is appropriate to consider that the allele frequencies reported for the European population are T (T) = ~60% and C (t) = ~40% (Table 2) while those reported for populations of East Asia correspond to T (T) = ~94% and C (t) = ~6% (https://www.ensembl.org (accessed on 30 March 2022)). Nonetheless, the pathology is widespread and has a significant impact on the quality of life, and although today the molecular methods useful to identify gene variants and bacterial strains are easily available, an increasing number of studies have assessed the correlation between gene polymorphisms and the presence or amounts of specific periodontopathogens. In this regard, a very recent and detailed systematic review of evidence of associations between host genetic variants and the detection and counting of periodontal microbes highlights in the literature only 19 articles on this topic, of which only three relate to polymorphisms in the VDR gene. A meta-analysis applied to these studies showed no association between SNPs within the IL10, IL6, IL4, IL8, IL17A, and VDR genes and periodontal pathogenic bacteria. However, the authors state that to date there is still a paucity of well-conducted case-control studies in periodontal infectogenomics [8,10]. In particular, Borges M.A.T. et al. (2009) examined 38 bacteria species in 30 patients with chronic periodontitis and 30 healthy Brazilian controls searching for a correlation with the *TaqI* variant [32]. The study did not demonstrate any association with the levels of the subgingival microbiota but indicated only a prevalence of the *TaqI* genotypes TC (Tt) and TT (TT) in the periodontitis group [33]. Similarly, Lauritano et al. found no association between the amounts of periodontal “red complex” species *P. gingivalis*, *T. forsythia*, and *T. denticola* and the *TaqI* variant as well as genetic polymorphisms of IL6 and IL10 in 326 Italian patients diagnosed with chronic periodontitis [34]. More recently, still in an Italian population of 96 CP, no correlations were found between VDR polymorphisms and a set of bacteria known to be involved in CP, identified and quantified by RT PCR [34]. After performing detection and quantification of *A. actinomycetemcomitans*, *P. gingivalis*, *T. forsythia*, *T. denticola*, and *P. intermedia* in a population of 1460 Thai subjects, Torrungruang et al. showed that subjects carrying the FokI CC + CT (FF + Ff) genotypes had greater *P. gingivalis* load and more severe periodontitis, compared to individuals with the TT genotype [36]. In our study, we found an association between FokI TT + CT (ff + Ff) genotypes and high levels of all quantized bacterial species except for *F. nucleatum* (Table 7 and Appendix A). The Fok1 polymorphism is characterized by a nucleotide substitution T→ C (c.2T > C) at the first codon of the start codon of the gene resulting in the replacement of methionine (ATG) with threonine (ACT) at amino acid position 1 (p.Met1Thr). This variant determines the creation of a new start codon at three amino acids downstream from the site of initiation of translation, with consequent alteration of the related protein. The C (F) variant therefore induces the synthesis of a shorter protein with greater biological activity and more effective in the transcriptional activity of the vitamin D signal [31,35]. The TT + CT+ (ff +Ff) genotypes are associated with lower bone mineral density than the CC (FF) genotype, as available on the Pharmacogenomics Knowledgebase (PharmGKB) (PharmGKB ID: 769164470) [54] and confirmed by a recent meta-analysis published this year conducted on 14 studies with 2219 participants [32]. In vitro studies have also shown that the shortest protein encoded by the C (F) allele is correlated with a more active immune response by increased transcriptional activity of NF-κB and NFAT and increased production of IL-12p40 [36,55]. These data are in line with a very recent study conducted in a Chinese cohort of 576 sepsis patients and 421 healthy controls showing that low vitamin D level and TT + CT (ff + Ff) genotypes were significantly more prevalent in sepsis patients [56]. Finally, our results indicate a correlation between high levels of *F. nucleatum* and a higher prevalence of the G (b) BsmI allele which, as we previously reported, is expected to modify the stability of mRNA with consequent effects on protein expression [26,32]. Oral pathologies, including periodontitis, are closely related to the nutritional habit and to the oral microbiota composition [22,35]. Although bacteria from the oral microbiota have been extensively studied, in recent years the role of fungi and viruses in oral pathology has been highlighted and even more so when associated with systemic comorbidities [17,57,58]. Gram (-) bacteria are isolated from all surfaces of the oral cavity [59,60,61]. They are involved in the formation of dental plaque and some species contribute to the development of caries [62,63]. Many of these are pathogens, such as the *Porphyromonas gingivalis*, which is an anaerobic Gram-negative bacterium that can cause various diseases such as periodontitis, abscesses, and endocarditis, and *T. denticola*, which is a Gram-negative, obligate anaerobic causing periodontitis [64,65,66]. Studies have shown that cotinine (a substance found in cigarette smoke) can interfere with *P. gingivalis* ability to bind and invade epithelial cells [67,68,69]. It is noteworthy that certain miRNAs, such as miRNA515-5p for *F. nucleatum*, have been demonstrated to be able to enter bacterial cells and induce gene expression, facilitating bacterial growth and progression of tissue damages [70,71]. This capability is very important for the bacterial pathogenicity, and *F. nucleatum* is currently under consideration for its ability to induce local and distant damages, including cancers [70,72,73,74]. In fact, it has been positively related to periodontitis but also to pancreatic and colic cancers [75]. Taken together, periodontal diseases may be a plausible risk factor for several diseases, including cancer, as observed at the earlier stages of disease, and ophthalmological disorders such as Pterygium, Age-related Macular Degeneration (AMD), and Diabetic Retinopathy (DR) [28,29,30,56]. Further studies should be encouraged for better understanding of this potential new relationship. The main strength of this pilot study is to have addressed a topic poorly described in the literature, with aspects still unclear and unknown and that could have important developments for the diagnosis, prognosis, and therapy of periodontal disease. The limitations are the low number of patients recruited for which we cannot exclude possible artifacts due to the numerosity, and so we believe that it is necessary in the future to expand the study population, providing a larger replication study to better sustain these statistically convincing results. Therefore, we recognize the need to expand the patient sample in future studies, hoping for larger replication studies.

## 5. Conclusions

In conclusion, our findings suggest the involvement of the VDR gene BsmI and TaqI polymorphisms in periodontal disease, while FokI and BsmI may be important in determining an increased presence of periodontopathogens. Confirmation of these data, for which further studies are needed, would better clarify the pathophysiological aspects of the disease, allow a better stratification of patients in specific risk groups on the basis of genetic susceptibility, and open to new potential perspectives for alternative therapeutic approaches. In fact, these data, once confirmed and validated, could suggest a potential use of specific molecular tests to improve the diagnosis of periodontal disease. Further development of our study could allow application in clinical practice of a personalized treatment for each individual patient in relation to different clinical presentations. In addition, it could give prognostic indications on the outcome of the disease.

## Figures and Tables

**Table 1 biomolecules-12-00833-t001:** Probe and primer sequences used for periodontal bacteria identification and quantization with real time PCR.

Bacterial Species	Primers and Probes	
*Internal Control Universal Bacterial*	Forward	TGGAGCATGTGGTTTAATTCGA
	Reverse	TGCGGGACTTAACCCAACA
	Probe	CACGAGCTGACGACA(AG)CCATGCA
*Aggregatibacter actinomycetemcomitans*	Forward	CAAGTGTGATTAGGTAGTTGGTGGG
	Reverse	CCTTCCTCATCACCGAAAGAA
	Probe	ATCGCTAGCTGGTCTGAGAGGATGGCC
*Porphyromonas gingivalis*	Forward	TGCAACTTGCCTTACAGAGGG
	Reverse	ACTCGTATCGCCCGTTATTC
	Probe	AGCTGTAAGATAGGCATGCGTCCCATTAGCTA
*Porphyromonas endodontalis*	Forward	TCCTACGGGAGGCAGCAGT
	Reverse	GGACTACCAGGGTATCTAATCCTGTT
	Probe	CGTATTACCGCGGCTGCTGGCAC
*Treponema denticola*	Forward	GGTCTTCTTATGGGTGCTGGGA
	Reverse	CTTCATATTCGCCCGTTGT
	Probe	GGTCTTCTTATGGGTGCTGTTA
*Tannerella forsythia*	Forward	GACAACCGGATCAGCGAAAT
	Reverse	TCATTGACTTGGCGGATCG
	Probe	TCAAATTGACACCGGCAACTACGTATAACTCGT
*Prevotella intermedia*	Forward	CCACATATGGCATCTGACGTG
	Reverse	TCAATCTGCACGCTACTTGG
	Probe	ACCAAAGATTCATCGGTGGAGGATGGG
*Fusobacterium nucleatum*	Forward	CAACCAT TACT T TAACTCTACCATGTTCA
	Reverse	GTTGACTTTACAGAAGGAGATTA TGTAAAAATC
	Probe	GTTGACTTTACAGA AGGAGATTATGTAAAAATC

**Table 2 biomolecules-12-00833-t002:** Comparison between allele frequencies obtained from the genotyping of our study and the allele frequency data available from the 1000 Genomes and GnomAD databases.

SNPs	Alleles	1000 Genomes Project Phase 3		GnomAD Genomes r3.0		Total 91 Subjects %	*p* Value *
		Global	European	Global	European		
VDR FokI rs2228570	C (*F*)	*67.2*	*62.2*	*66.4*	61.7	**58.8**	*0.24*; *0.66*; *0.30*; *0.66*
	T (*f*)	*32.8*	*37.8*	*33.6*	38.3	**41.2**	
VDR BsmI rs1544410	A (*B*)	*29.6*	*40.4*	*35.1*	40.2	**20.3**	*0.13*; ***0.001***; *0.017*; ***0.001***
	G (*b*)	*70.4*	*59.6*	*64.9*	59.8	**79.7**	
VDR ApaI rs7975232	A (*A*)	*51.5*	*55.5*	*55.3*	*52.7*	** *43.4* **	*0.22*; *0.07*; *0.08*; *0.15*
	C (*a*)	*48.5*	*44.5*	*44.7*	*47.3*	**56.6**	
VDR TaqI rs731236	T (*T*)	*72.3*	*60.0*	*66.1*	60.4	**80.2**	*0.18*; ***0.002***; ***0.002***; ***0.002***
	C (*t*)	*27.7*	*40.0*	*33.9*	39.6	**19.8**	

* *p* values relative to the comparison with 1000 Genomes Project Phase 3 Global and European population and gnomAD genomes r3.0 global and European population respectively.

**Table 3 biomolecules-12-00833-t003:** Distributions of genotype and allele frequencies of SNPs VDR FokI (Rs2228570), VDR BsmI (Rs1577710), VDR ApaI (Rs7975232), and VDR TaqI (Rs731236) in health controls and periodontitis patients.

			Health Controls (*n* = 41)	Periodontitis pts (*n* = 50)	*p* Value
**VDR FokI** **rs2228570** Exon2 c.2T > C (*f > F*) * *p.Met1Thr*	Genotypes (%)	C/C (F/F)	17 (*41*)	14 (*28*)	0.19
		C/T (*F/f*)	20 (*49*)	25 (*50*)	
		T/T (*f/f*)	4 (*10*)	11 (*22*)	
	Alleles (%)	C (*F*)	54 (*66*)	53 (*53*)	0.079
		T (*f*)	28 (*34*)	47 (*47*)	
	HW (p)		0.74	1	
**VDR BsmI** **rs1544410** Intron 8 c.1024 + 283G > A(*b > B*) *	Genotypes (%)	A/A (*B/B*)	3 (*7*)	1 (*2*)	
		A/G (*B/b*)	21 (*51*)	8 (*16*)	**0.0001**
		G/G (*b/b*)	17 (*41*)	41 (*82*)	
	Alleles (%)	A (*B*)	27 (*33*)	10 (*10*)	**0.0003**
		G (*b*)	55 (*67*)	90 (*90*)	
	HW (p)		0.48	0.39	
**VDR ApaI** **rs7975232** Intron 8 c.1025−49A > C(*A > c*) *	Genotypes (%)	A/A (*A/A*)	10 (*24*)	5 (*10*)	
		A/C (*A/a*)	23 (*56*)	26 (*52*)	0.064
		C/C (*a/a*)	8 (*20*)	19 (*38*)	
	Alleles (%)	A (*A*)	43 (*52*)	36 (*36*)	**0.026**
		C (*a*)	39 (*48*)	64 (*64*)	
	HW (p)		0.54	0.54	
**VDR TaqI** **rs731236** Exon 9 c.1056T > C (*T > t*) p.Ile352=	Genotypes (%)	T/T (*T/T*)	15 (*37*)	45 (*90*)	
		T/C (*T/t*)	21 (*51*)	5 (*10*)	**<0.00001**
		C/C (*t/tT*)	5 (*12*)	0 (*0*)	
	Alleles (%)	T (*T*)	51 (*62*)	95 (*95*)	**<0.00001**
		C (*t*)	31 (*38*)	5 (*5*)	
	HW (p)		0.74	1	

HWE: Hardy–Weinberg equilibrium, significant difference was accepted at *p* < 0.05. * Correspondence of nomenclature of SNP alleles.

**Table 4 biomolecules-12-00833-t004:** Haplotype analysis performed on SNPs VDR FokI (Rs2228570), VDR BsmI (Rs1577710), VDR ApaI (Rs7975232), and VDR TaqI (Rs731236) and their corresponding frequencies in health controls (*n* = 41) and periodontitis patients (*n* = 50) patients.

Haplotypes				Frequency			χ^2^	*p* Value	*Odds Ratio [95% CI]*
FokI	BsmI	ApaI	TaqI	Total	Health Controls	Patients			
**C** (*F*)	**G** (*b*)	**C** (*a*)	**T** (*T*)	*0.2179*	0.172	0.260	2.189	*0.139059*	*1.728 [0.833–3.583]*
**T** (f)	**G** (*b*)	**C** (*a*)	**T** (*T*)	*0.182*	0.0	0.288	28.586	** *0.0000000936* **	-
**C** (*F*)	**G** (*b*)	**A** (*A*)	**T** (*T*)	*0.158*	0.163	0.200	0.469	*0.493583*	*1.306 [0.607–2.811]*
**C** (*F*)	***A***(*B*)	**C** (*a*)	**T** (*T*)	*0.0995*	0.111	0.050	2.277	*0.131400*	*0.425 [0.136–1.326]*
**T** (f)	**G** (*b*)	**A** (*A*)	**T** (*T*)	*0.0964*	0.057	0.112	1.814	*0.178070*	*2.131 [0.694–6.543]*
**C** (*F*)	**G** (*b*)	**A** (*A*)	**C** (*t*)	*0.0806*	0.109	0.0	10.351	** *0.001302* **	*0.000 [0.000–0.002]*
**T** (*f*)	**A** (*B*)	**A** (*A*)	**C** (*t*)	*0.0425*	0.100	0.0	10.304	** *0.001335* **	*-*
**T** (*f*)	**A** (*B*)	**A** (*A*)	**T** (*T*)	*0.0334*	0.049	0.0	4.921	** *0.026577* **	*-*
**C** (*F*)	**G** (*b*)	**C** (*a*)	**C** (*t*)	*0.0319*	0.104	0.0	10.749	** *0.001051* **	*-*
**T** (*f*)	**G** (*b*)	**A**(*A*)	**C**(*t*)	*0.0233*	0.047	0.027	0.475	*0.490768*	*0.578 [0.119–2.798]*
**T** (*f*)	**A** (*B*)	**C** (*a*)	**T** (*T*)	*0.0151*	0.070	0.030	1.534	*0.215636*	*0.415 [0.099–1.736]*

**Table 5 biomolecules-12-00833-t005:** Linkage disequilibrium coefficient (D’) of VDR SNPs typed in this study.

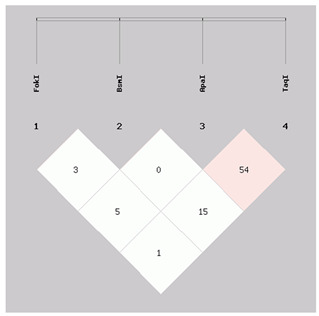			
D’	VDR BsmI rs1544410	VDR ApaI rs7975232	VDR TaqI rs731236
VDR FokI rs2228570	0.033	0.058	0.013
VDR BsmI rs1544410	-	0.009	0.150
VDR ApaI rs7975232	-	-	0.545

**Table 6 biomolecules-12-00833-t006:** Results of the bacterial load by real-time PCR from health controls and periodontitis patients’ samples.

	Aggregatibacter Actinomycetemcomitans	Porphyromonas Gingivalis	Porphyromonas Endodontalis	Treponema Denticola	Tannerella Forsythia	Prevotella Intermedia	Fusobacter Nucleatum
**Health controls (*n* = 41)**							
Media *	25	243	859	245	305	165	17,047
SD	*17*	*168*	*394*	*202*	*210*	*134*	*13,540*
Range *	*0–76*	*32–744*	*321–1940*	*87–1200*	*87–980*	*34–543*	*1340–45,570*
**Periodontitis pts (N = 50)**							
Media *	264,331	3,963,445	3,962,550	373,472	363,527	1,466,116	936,526
SD	*775,494*	*14,409,905*	*12,856,465*	*1,251,967*	*1,024,361*	*8,374*	*141,000*
Range *	*0–4,950,000*	*256–78,100,000*	*351–83,500,000*	*0–6,300,000*	*0–6.200.000*	*542–31,800,000*	*0–14,000,000*
**Cut off used to divide patients into 2 groups**	100,000	100,000	15,000	10,000	100,000	100,000	100,000
**Patients < cut off (Low)**	36	28	26	22	33	27	21
**Patients > cut off (High)**	14	22	24	28	17	23	29

* Values expressed in copies/mL.

**Table 7 biomolecules-12-00833-t007:** Distributions of genotype frequencies of VDR SNPs that had a significant correlation comparing patients with low (LOW) and high (HIGH) levels of bacterial load and corresponding ORs and 95% CI.

Bacteria	SNP	Genotype	Low (%)	High (%)	*p* Value	Model	Genotype	OR (95% CI)	*p* Value
Aggregatibacter Actinomycetemcomitans Patients LOW N = 36 Patients HIGH N = 14	**FokI** **rs2228570**	C/C (*F/F*)	13 (36)	1 (7)	0.034708	Codominant	C/T	0.10 (0.01–0.87)	0.025
		C/T (*F/f*)	14 (39)	11 (79)		Dominant	C/T-T/T	0.14 (0.02–1.16)	0.025
		T/T (f/f)	9 (25)	2 (14)		Overdominant	C/T	0.17 (0.004–0.73)	0.0099
Porphyromonas Gingivalis Patients LOW N = 28 Patients HIGH N = 22	**FokI** **rs2228570**	C/C (*F/F*)	12 (43)	2 (9)	0.022173	Codominant	C/T	0.11 (0.02–0.61)	0.016
		C/T (*F/f*)	10 (36)	15 (68)		Dominant	C/T-T/T	0.13 (0.03–0.68)	0.0057
		T/T (*f/f*)	6 (21)	5 (23)		Overdominant	C/T	0.26 (0.08–0.85)	0.021
Porphyromonas endodontalis Patients LOW N = 26 Patients HIGH N = 24	**FokI** **rs2228570**	C/C (*F/F*)	12 (46)	2 (8)	0.010462	Codominant	C/T	0.09 (0.02–0.52)	0.007
		C/T (*F/f*)	9 (35)	16 (67)			T/T	0.14 (0.02–0.94)	
		T/T (*f/f*)	5 (19)	6 (25)		Dominant	C/T-T/T	0.11 (0.02–0.55)	0.0019
						Overdominant	C/T	0.26 (0.08–0.85)	0.022
Treponema Denticola Patients LOW N = 22 Patients HIGH N = 28	**FokI** **rs2228570**	C/C (*F/F*)	10 (45)	4 (14)	0.045747	Codominat	C/T	0.23 (0.05–0.77)	0.043
		C/T (*F/f*)	9 (41)	16 (57)			T/T	0.15 (0.03–0.87)	
		T/T (*f/f*)	3 (14)	8 (29)		Dominant	C/T-T/T	0.20 (0.05–0.77)	0.014
Tannerella Forsythia Patients LOW N = 33 Patients HIGH N = 17	**FokI** **rs2228570**	C/C (*F/F*)	13 (39)	1 (6)	0.034969	Codominat	C/T	0.08 (0.01–0.74)	0.02
		C/T (*F/f*)	13 (39)	12 (71)		Dominant	C/T-T/T	0.10 (0.01–0.82)	0.0064
		T/T (*f/f*)	7 (21)	4 (24)		Overdominant	C/T	0.27 (0.08–0.95)	0.035
Prevotella Intermedia Patients LOW N = 27 Patients HIGH N = 23	**FokI** **rs2228570**	C/C (*F/F*)	13 (48)	1 (4)	0.002382	Codominat	C/T T/T	0.04 (0.00–0.39) 0.06 (0.01–0.42)	0.0009
		C/T (*F/f*)	9 (33)	16 (70)		Dominant	C/T-T/T	0.05 (0.01–0.42)	0.0002
		T/T (*f/f*)	5 (19)	6 (26)		Overdominant	C/T	0.22 (0.07–0.72)	0.0098
Fusobacter Nucleatum Patients LOW N = 21 Patients HIGH N = 29	**BsmI** **rs1544410**	A/A (*B/B*)	1 (5)	0 (0)	0.049991	Codominat	A/G	5.79 (1.03–32.49)	0.041
		A/G (*B/b*)	6 (29)	2 (7)		Dominant	A/G-A/A	6.75 (1.23–36.91)	0.016
		G/G (*b/b*)	14 (67)	27 (93)		Overdominant	A/G	5.40 (0.97–30.17)	0.038

## Data Availability

Data available upon request from the corresponding authors. All data presented in this study are available.

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
