# Peer review of "Association between Vitamin D Receptor Gene Polymorphisms and Periodontal Bacteria: A Clinical Pilot Study"

_biomolecules, 2022, doi:10.3390/biom12060833_

Round 1

Reviewer 1 Report

A very nicely carried out study to examine the associations between SNPs in the VDR gene and periodontal disease. While it doesn’t break new ground, it makes a good contribution to the overall field. The main issue is with the discussion. While a good amount of discussion was provided to put their results in context with the prior studies in the field, there should be some discussion of the possible mechanism by which these SNPs can lead to susceptibility to periodontal disease. There are a number of publications that should be brought into the discussion that relate vitamin D mechanistically to periodontal disease and overall oral health, both in vitro and in animal models. Incorporation of these studies will greatly enhance the significance of this study.

Author Response

Reviewer 1

A very nicely carried out study to examine the associations between SNPs in the VDR gene and periodontal disease. While it doesn’t break new ground, it makes a good contribution to the overall field. The main issue is with the discussion. While a good amount of discussion was provided to put their results in context with the prior studies in the field, there should be some discussion of the possible mechanism by which these SNPs can lead to susceptibility to periodontal disease. There are a number of publications that should be brought into the discussion that relate vitamin D mechanistically to periodontal disease and overall oral health, both in vitro and in animal models. Incorporation of these studies will greatly enhance the significance of this study.

Author response: We thank the reviewer for his valuable comments. According with the reviewer’s suggestions the text has been implementeddescribing the suggested topics with the appropriate references.

Reviewer 2 Report

This study entitled "Association between vitamin D receptor gene polymorphisms (FokI, BsmI, ApaI and TaqI) and periodontal bacteria: a Pilot Study" has merits and it is of interest for the readers of biomolecules. 

To improve the paper, please, follow these suggestions:

1. Modify the title in "Association between vitamin D receptor gene polymorphisms and bacteria: a clinical pilot study."

2. The aim of the study should be better described both in the abstract and in the main text.

3. Describe the state of the art of the most innovative technique to manage periodontal inflammation  (as an example: Gardin, C. et al. Hyperbaric Oxygen Therapy Improves the Osteogenic and Vasculogenic Properties of Mesenchymal Stem Cells in the Presence of Inflammation In Vitro. Int J Mol Sci.)

4. Improve the conclusions and add the limitations of this study.

5. Check the format of references, as some typos are present.

Author Response

Reviewer 2

This study entitled "Association between vitamin D receptor gene polymorphisms (FokI, BsmI, ApaI and TaqI) and periodontal bacteria: a Pilot Study" has merits and it is of interest for the readers of biomolecules. 

To improve the paper, please, follow these suggestions:

  1. Modify the title in "Association between vitamin D receptor gene polymorphisms and bacteria: a clinical pilot study."

Author response:  We thank the reviewer for his valuable comments and constructive criticism that can improve the quality of our paper. According with the suggestion we changed the title of the paper.

  1. The aim of the study should be better described both in the abstract and in the main text.

Author response: According with the suggestion the text was modified by more accurately specifying the aim of the study.

  1. Describe the state of the art of the most innovative technique to manage periodontal inflammation(as an example:Gardin, C. et al. Hyperbaric Oxygen Therapy Improves the Osteogenic and Vasculogenic Properties of Mesenchymal Stem Cells in the Presence of Inflammation In Vitro.Int J Mol Sci.)

Author response:  According with the suggestion the text has been expanded by introducing the most innovative methodologies to manage periodontal inflammation

  1. Improve the conclusions and add the limitations of this study.

Author response:  The text has been modified as suggested by the reviewer

  1. Check the format of references, as some typos are present.

Author response:  Typos have been corrected as suggested by the reviewer

Reviewer 3 Report

01

“We selected 50 unrelated Caucasian patients (26 males and 24 females; mean age 47 ± 8 yrs; range 21-63 yrs) affected by periodontal disease”

The selection of the patients was not made randomly, which introduces bias in the selection process.

02

No power analysis was performed. Therefore, it is not possible to know whether the analysis of the results of the present study is a true finding or a pure chance. This may compromise the entire validity of this study.

03

“Data were analysed by using Student’s t-test or one-way ANOVA with Bonferroni post-test, as appropriate.”

The authors opted for parametric tests from the start, even though no normality and homoscedasticity were checked.

04

“Taken together, periodontal diseases may be a plausible risk factor for several diseases, including cancer, as observed at the earlier stages of disease, and ophthalmological disorders such as Pterygium, Age-related Macular Degeneration (AMD) and Diabetic Retinopathy (DR) [28–30,56]”

This is not a conclusion of your work. This is based on the literature, and should stay in the Introduction.

05

“Further studies should be encouraged for better understanding of this potential new relationship.”

“Therefore, we recognize the need to expand the patient sample in future studies, hoping for larger replication studies.”

“Confirmation of these data, for which further studies are needed, would better clarify the pathophysiological aspects of the disease, allow a better stratification of patients on the basis of genetic susceptibility and open to new potential perspectives for alternative therapeutic approaches.”

This are not conclusions of your work. These are recommendations/suggestions/opinions, and should stay in the Discussion.

06

“The main strength of this pilot study is to have addressed a topic poorly described in the literature, with aspects still unclear and unknown and that could have important developments for the diagnosis, prognosis and therapy of periodontal disease. The limitations are the low number of patients recruited for which we cannot exclude possible artifacts due to the numerosity.”

This is not a conclusion of your work. This consists of strengths/limitations of the study, and should stay in the Discussion.

07

There are some sentences in the text without reference to a previous study (or studies) in order to give evidence to their statements. Without references, these statements would be mere assumptions or allegations made by the author. Therefore, each of the following sentences need at least one reference to back up their statement:

“Gram (-) bacteria are isolated from all surfaces of the oral cavity. They are involved in the formation of dental plaque and some species contribute to the development of caries. Many of these are pathogens, such as the Porphyromonas gingivalis, which is an anaerobic Gram-negative bacterium that can cause various diseases such as periodontitis, abscesses and endocarditis, and T. denticola, which is a Gram-negative, obligate anaerobic causing periodontitis. Studies have shown that cotinin (a substance found in cigarette smoke) can interfere with P. gingivalis ability to bind and invade epithelial cells. It is noteworthy that certain miRNAs, such as miR-NA515-5p for F. nucleatum, have been demonstrated to be able to enter bacterial cells and induce gene expression, facilitating bacterial growth and progression of tissue damages. This capability is very important for the bacterial pathogenicity, and F. nucleatum is currently under consideration for its ability to induce local and distant damages, including cancers.”

08

“a few studies have assessed the correlation between gene polymorphisms and the presence or amounts of specific periodontopathogens.”

“highlights in the literature only 19 articles on this topic.”

Since when are 19 studies “only” “few studies”?

Author Response

Reviewer 3

01

“We selected 50 unrelated Caucasian patients (26 males and 24 females; mean age 47 ± 8 yrs; range 21-63 yrs) affected by periodontal disease”

The selection of the patients was not made randomly, which introduces bias in the selection process.

Author response: We thank the reviewer for his valuable commentsand constructive criticism that can improve the quality of our paper. This study is an observational, longitudinal, case-control and single time study where the enrollment was performed considering the inclusion criteria. We do not believe in the presence of a bias in the selection process, as selection occurred casually, with no selection for the analysis and the visit times were not related to the outcome of interest.

02

No power analysis was performed. Therefore, it is not possible to know whether the analysis of the results of the present study is a true finding or a pure chance. This may compromise the entire validity of this study.

Author response: We apologize for missing the description of Power analysis that was performed at beginning of the study and calculated by using the GPower software in the presence of the following parameters: effect size d=0.5 (moderate) and α error=0.05 for evaluating the reliability of the results with a power of at least 0.70 value, that provided a case-control of 51-51 for a total of 102 subjects in the study population (see panels in A). Since the real study population was 50 cases and 41 controls, the effective power of the study was recalculated showing a value of 0.7 (see panels in B) (see ref 53 Hedges LV, Pigott TD. The power of statistical tests in meta-analysis. Psychol Methods 2001;6:203–17). Therefore, the phrase “A formal sample size calculation was performed at beginning with the following parameters: effect size d = 0.5 (medium); α error probability = 0.05 and an allocation ratio of 1 (Genetic Power Calculator G Power 3.1.9.4; free available) [53]. The real power of this study population, for a total size of 91 subjects, is 0.76, considering means from these two independent groups (50 case and 41 controls)⁠.” was introduced in the text. Furthermore, we highlight that at least 12 participants are required for a pilot study with primary focus of estimating average values and variability for planning larger subsequent studies.

Below the protocol of formal (A) and effective (B) power analysis with distribution (B):

A

B

03

“Data were analysed by using Student’s t-test or one-way ANOVA with Bonferroni post-test, as appropriate.”

The authors opted for parametric tests from the start, even though no normality and homoscedasticity were checked.

Author response: We apologize for missing some important parts in the statistical analysis paragraph. We performed a normality test according to a standard initial statistical analysis: Row values were analyzed by the Shapiro-Wilk tests to satisfy the assumption of data from a normally distributed population. As we obtained a pValue p>0.05, we opted for parametric tests. As well, a homoscedasticity check was also performed using the F test. The following phrase “Row values were analyzed by the Shapiro-Wilk tests and F test to satisfy normality and variance assumptions.” was introduced in 2.5 paragraph of the tracked-ON revised manuscript.

04

“Taken together, periodontal diseases may be a plausible risk factor for several diseases, including cancer, as observed at the earlier stages of disease, and ophthalmological disorders such as Pterygium, Age-related Macular Degeneration (AMD) and Diabetic Retinopathy (DR) [28–30,56]”

This is not a conclusion of your work. This is based on the literature, and should stay in the Introduction.

Author response: We agree with the reviewer, this is not a conclusion of our study but a possible suggestion to investigate in other diseases were periodontitis represents a common comorbidity. It is also well-described in literature that in some cases the ophthalmologis can suggest a dentist observation. This is actually the base of multidisciplinarity applied to the Sanitary Service, increasing specialist courses are actually available (endocrinology-ginecology and ophthalmology; neurology-endocrinology-psychiatry-ophthalmology, … and so on). Dentist-oncology-genetics and herein we prospect the future Dental-genetic-ophthalmologic and geriatric route).

05

“Further studies should be encouraged for better understanding of this potential new relationship.”

“Therefore, we recognize the need to expand the patient sample in future studies, hoping for larger replication studies.”

“Confirmation of these data, for which further studies are needed, would better clarify the pathophysiological aspects of the disease, allow a better stratification of patients on the basis of genetic susceptibility and open to new potential perspectives for alternative therapeutic approaches.”

This are not conclusions of your work. These are recommendations/suggestions/opinions, and should stay in the Discussion.

Author response: The text has been modified as suggested by the reviewer

06

“The main strength of this pilot study is to have addressed a topic poorly described in the literature, with aspects still unclear and unknown and that could have important developments for the diagnosis, prognosis and therapy of periodontal disease. The limitations are the low number of patients recruited for which we cannot exclude possible artifacts due to the numerosity.”

This is not a conclusion of your work. This consists of strengths/limitations of the study, and should stay in the Discussion.

Author response:  The text has been modified as suggested by the reviewer

07

There are some sentences in the text without reference to a previous study (or studies) in order to give evidence to their statements. Without references, these statements would be mere assumptions or allegations made by the author. Therefore, each of the following sentences need at least one reference to back up their statement:

“Gram (-) bacteria are isolated from all surfaces of the oral cavity. They are involved in the formation of dental plaque and some species contribute to the development of caries. Many of these are pathogens, such as the Porphyromonas gingivalis, which is an anaerobic Gram-negative bacterium that can cause various diseases such as periodontitis, abscesses and endocarditis, and T. denticola, which is a Gram-negative, obligate anaerobic causing periodontitis. Studies have shown that cotinin (a substance found in cigarette smoke) can interfere with P. Gingivalis ability to bind and invade epithelial cells. It is noteworthy that certain miRNAs, such as miR-NA515-5p for F. nucleatum, have been demonstrated to be able to enter bacterial cells and induce gene expression, facilitating bacterial growth and progression of tissue damages. This capability is very important for the bacterial pathogenicity, and F. Nucleatum is currently under consideration for its ability to induce local and distant damages, including cancers.”

Author response: According with the suggestions we added the appropriate references

08

“a few studies have assessed the correlation between gene polymorphisms and the presence or amounts of specific periodontopathogens.”

“highlights in the literature only 19 articles on this topic.”

Since when are 19 studies “only” “few studies”?

Author response:  We apologize for the misunderstanding that occurred and the lack of clarity in the sentence. Indeed we agree that 19 studies of correlation between gene polymorphisms and the presence or amounts of specific periodontopathogens are not so few. We correct “few studies” with “increasing studies”. However, it is necessary to specify that of the 19 studies mentioned only 3 refer to VDR gene polymorphisms while the remaining studies are related to polymorphisms of other genes such as IL10, IL6, IL4, IL8 or IL17 (Reference 8: Zoheir N.et al Periodontal Infectogenomics: A Systematic Review Update of Associations between Host Genetic Variants and Subgingival Microbial Detection. Clin. Oral Investing. 2022, 26, 2209–2221, doi:10.1007/S00784-021-04233-8). Therefore, in accordance with the reviewer's comments, we have changed the text to better specify the concept in detail.

Round 2

Reviewer 2 Report

none to add 

Reviewer 3 Report

The manuscript now seems to be suitable for publication.